# Post-Polio Syndrome: Impact of Humoral Immune Deficiencies, Poliovirus Neutralizing Antibodies, Vitamin D Deficiency

**DOI:** 10.3390/vaccines13090939

**Published:** 2025-09-02

**Authors:** Antonio Toniolo, Konstantin Chumakov, Giovanni Federico, Giuseppe Maccari, Angelo Genoni, Alessandro Saba, Andrea Nauti, Giorgio Bono, Franco Molteni, Salvatore Monaco

**Affiliations:** 1Global Virus Network, University of Insubria, 21100 Varese, Italy; 2Department of Microbiology, Immunology, and Tropical Medicine, George Washington University, Washington, DC 20052, USA; kostyachumakov@me.com; 3Unit of Pediatrics, Department of Clinical and Experimental Medicine, University of Pisa, 56126 Pisa, Italy; giovanni.federico@unipi.it; 4Data Science for Health Lab, Fondazione Toscana Life Sciences, 53100 Siena, Italy; g.maccari@toscanalifesciences.org; 5Medical Microbiology Section, Department of Medicine, University of Insubria, 21100 Varese, Italy; angelopaolo.genoni@uninsubria.it; 6Department of Surgical, Medical and Molecular Pathology, University of Pisa, 56126 Pisa, Italy; alessandro.saba@unipi.it; 7Clinical Laboratories, Ospedale di Circolo e Fondazione Macchi, 21100 Varese, Italy; andrea.nauti@asst-settelaghi.it; 8Neurology Section, Department of Biotechnology, University of Insubria, 21100 Varese, Italy; giorgiogiovannibono@gmail.com; 9Villa Beretta Rehabilitation Research Innovation Institute, 23845 Costa Masnaga, Italy; francomolteni@vbrrii.it; 10Department of Neurosciences, University of Verona and Policlinico GB Rossi, 37134 Verona, Italy

**Keywords:** post-polio syndrome, poliovirus, virus susceptibility, antibody deficiency, neutralizing antibodies, vitamin D, polio vaccine

## Abstract

**Background/Objectives**: This study investigated susceptibility factors that may contribute to Post-Polio Syndrome (PPS) in elderly polio survivors. **Methods**: Serum immunoglobulin (Ig) levels, poliovirus neutralizing antibodies (PV NAb), and vitamin D status were evaluated in 80 PPS patients, 40 family members, and 89 healthy controls. **Results**: A significant number of PPS patients and their family members showed reduced levels of total IgG and/or IgA, and specific IgG subclasses, indicating a high prevalence of primary humoral immunodeficiencies within these groups. Despite these Ig deficits, PV NAb titers were similar across all groups, indicating high protection against poliovirus, likely due to vaccination campaigns with live virus in Italy and intense exposure to poliovirus, especially in long-term rehabilitation institutions. However, a small group of PPS subjects lacked neutralizing antibodies for specific poliovirus serotypes, suggesting more severe antibody deficiencies. Additionally, PPS subjects had a high prevalence of vitamin D deficiency, which likely increases their risk for osteoporosis/osteopenia and fractures. It is unclear if this deficiency was also present in their infancy, potentially enhancing their susceptibility to poliovirus. **Conclusions**: Overall, the findings indicate that genetic, immunological, or nutritional factors may increase individual susceptibility to the pathogenic effects of poliovirus. This study—limited to serum antibodies—highlights the complex relationship between immune status and long-term health in aging polio survivors. The results emphasize the need for potent poliovirus drugs and vaccines to help contain possible outbreaks but also—for poliomyelitis survivors—to avoid or mitigate the progression to PPS, the latest phase of this devastating disease.

## 1. Introduction

Polioviruses (PVs) belong to the C species of the Enterovirus (EV) genus and comprise three different types (1 to 3). During the pre-vaccine era, approximately 1 in 190 infections with PV type 1 resulted in paralytic poliomyelitis (PPM), a substantially higher incidence compared to PV-2 or PV-3 infections, which led to PPM in only 1 in 1100 to 1900 individuals [1]. Despite the understanding of infectivity-to-paralysis ratios, the precise mechanisms governing susceptibility to PV infections and the development of PPM remain unclear. Research has explored various predisposing factors, including host genetic and immunological elements, alongside variables such as age at infection, hygienic conditions, lactation practices, and nutritional status.

Susceptibility to infectious agents is influenced by the host’s genetic background [2]. Accordingly, studies have suggested that specific genetic factors may modulate the outcome of enteroviral diseases. Specifically, HLA haplotypes could contribute to PPM resistance [3] and rare mutations in the PV receptor CD155 may favor the development of PPM [4]. In addition, single-nucleotide polymorphisms (SNPs) in TLR3 and TLR7 may confer susceptibility to severe forms of EV-induced hand, foot, and mouth disease (HFMD) [2] and variants of the EV-A71 receptor PSGL-1 may predispose to severe HFMD. Despite these findings, no single genetic factor is believed to solely regulate susceptibility to PVs.

Immunologically, inborn errors of immunity (IEIs) are associated with enhanced susceptibility to infections [5] and, particularly, to EV infections [6]. Chronic shedding of vaccine-derived poliovirus (VDPV) occurs in individuals with defects in B-cell-mediated immunity. Most VDPV carriers present with a Predominantly Antibody Deficiency (PAD), but approximately one-fifth are diagnosed with a combined immunodeficiency [7]. Crucially, neutralizing secretory IgA plays a prominent role in limiting PV replication in mucosae of the digestive tract, while neutralizing IgG protects the central nervous system (CNS) from PV entry.

Long experience from polio vaccine campaigns underscores the need for implementation of comprehensive public health strategies that address malnutrition, limited access to clean water and inadequate sanitation. All these elements promote morbidity, mortality, and the transmission of viruses. The concurrent administration of vitamin A supplements has enhanced the efficacy of some polio vaccine campaigns [8,9]. Preclinical research in mice shows that co-administration of inactivated PV vaccine with vitamin D3 (cholecalciferol) enhances the systemic and mucosal antiviral immune responses [10]. Determined by diet and sunlight exposure (as well as by genetic background), an individual’s vitamin D status is best evaluated by measuring serum levels of 25-hydroxyvitamin D [11].

Vitamin D plays key roles in immunomodulation and bone health. Recently, it has been demonstrated that vitamin D deficiency/insufficiency was present in the majority of patients hospitalized with COVID-19 or influenza A, and that low levels correlate with disease severity and tend to persist indefinitely in survivors [12].

In humans, vitamin D supplementation modulates the immune transcriptome upregulating the presentation of antigens and viral response pathways. However, some immune signaling pathways are downregulated, including tumor necrosis factor-alpha and interleukin (IL)-17 [13] as well as the production in vitro of IL-2 and IFN-gamma [14]. It was also shown that vitamin D promotes the transition from pro-inflammatory IFN-gamma Th1 cells to suppressive IL-10+ cells [15].

Recent clinical trials demonstrated a protective activity of vitamin D against the progression of Multiple Sclerosis (MS) [16,17], an immuno-mediated disease of possible viral origin [18]. Thus, investigating the vitamin D status in people suffering severe infections may be of clinical relevance.

To advance the understanding of the individual susceptibility to paralytic poliomyelitis and to its long-term sequelae, we investigated aging polio survivors diagnosed with the Post-Polio Syndrome (PPS), a disorder coded as G14 in the International Classification of Diseases (ICD-10-CM) with the aim to assess potential immunoglobulin (Ig) deficiencies, poliovirus-neutralizing antibody (PV NAb) titers, and vitamin D levels.

## 2. Materials and Methods

### 2.1. Ethics

The study was approved by the Ospedale di Circolo & Fondazione Macchi, Comitato Etico dell’Insubria, Varese, Italy [REC: 0018898/2011; 0041619/2012; 003645/2013; 00811/2018—30 October 2018] and the Policlinico GB Rossi, Research Ethics Committee, 37124 Verona, Itay [Multicenter study NCT02176863/2014-2023—25 June 2014] [19]. Studies have been performed in accordance with the Declaration of Helsinki and its revisions. Before clinical assessment, all patients and controls signed informed written consent to medical procedures and studies of their biological samples.

### 2.2. Participating Hospitals, PPS Patients and Control Groups

A total of 209 individuals were investigated. Participants to the study were recruited between February 2011 and November 2019 at tertiary care Hospitals of Northern Italy specializing in Neurology and Rehabilitation: Policlinico GB Rossi, Verona, Italy; Villa Beretta Rehabilitation Center, Costa Masnaga, Italy; Ospedale di Circolo e Fondazione Macchi, Varese, Italy. Polio survivors had a verified diagnosis of poliomyelitis, supportive clinical data, imaging data, and electromyographic findings. Exclusion criteria: history of cerebrovascular disease, neuropathy from causes other than polio, congenital or acquired immune deficiencies, diabetes mellitus, drug or alcohol dependence. Blood samples were drawn in BD Vacutainer serum tubes after overnight fasting, either at the Ospedale di Circolo e Fondazione Macchi in Varese or procured at participating hospitals and shipped in dry ice to the laboratory in Varese. Samples were centrifuged at 4 °C, and supernatants were divided into aliquots and stored without delay at −80 °C. Control groups comprised (a) healthy family members of PPS subjects who had lived with them for at least five years (64% were siblings and offspring, 36% were partners or kindred), for whom the clinical conditions were evaluated by the same medical staff who authored this manuscript; (b) healthy controls (blood donors of 35 to 60 years of age). With some exceptions, in Italy people can donate blood from age 18 to 60. Blood samples from each PPS subject, his/her family member(s) and control blood donors were obtained on the same day. The number of subjects in each of the three groups was determined by the number of serum samples available for each assay. At the time the study was designed, it was impossible to predict precisely the number of recruitable cases and controls. Nonetheless, the study’s initial requirement of at least 28 participants per group was met for all assays. Table 1 summarizes the demographic and clinical data of the subjects investigated. Although the groups were not deliberately matched, their age, biological sex and socioeconomic status were suitable for comparison.

Polio survivors were admitted to participating hospitals because of progressive worsening of weakness and fatigue during daily and/or working activities. According to principles of the European Federation of Neurological Societies [20], polio survivors were classified as having PPS if they met the following criteria: (a) confirmed history of polio; (b) a period of partial or fairly complete functional recovery after the acute episode; (c) a period of at least 15 years of stable neurological function; (d) two or more of the following new symptoms after the stable period: new muscle weakness, muscle fatigue, new muscle atrophy, muscle and joint pain, breathing or swallowing problems, sleep disorders, cold intolerance; (e) exclusion of other neurological, medical and orthopedic problems. PPS is characterized by gradual progression over time [20,21]. Clinical features include new muscular weaknesses persisting for at least one year, fasciculations, atrophy of previously unaffected limbs, functional deficit of bulbar-innervated muscles, muscles of respiration, chronic fatigue, joint and muscle pain, cold intolerance, swallowing problems, sleep disorders, a reactive depressive syndrome.

### 2.3. Quantification in Serum of Major Ig Classes and IgG Subclasses

Blood was drawn in BD Vacutainer serum tubes, centrifuged within two hours and stored at −80 °C. According to manufacturers’ protocols, Ig antibodies of the three major classes were measured using the IgG, IgA, IgM kits of Beckman–Coulter (Cassina de’Pecchi, Italy) in conjunction with the IMMAGE Immunochemistry system and calibrators. Quantification of IgG subclasses was performed using the Peliclass IgG subclass Plus kit (Sanquin, Amsterdam, The Netherlands). The assay is based on nephelometry for IgG1 and IgG2 and latex turbidimetry for IgG3 and IgG4. The WHO 67/97 preparation was used as a reference. Calibrators had the following values: 6.210 g/L for IgG1, 3.450 g/L for IgG2, 0.390 g/L for IgG3, 0.591 g/L for IgG4. Regarding Ig levels, a guide to reference ranges for adult Caucasians has been used [22]. By means of the above reported methods, limits for significantly low Ig levels have been validated at our own laboratory (Ospedale di Circolo, Varese, Italy) in a cohort of 120 healthy adult blood donors (median age 51.2 yrs; IQR: 32.5–61.7 yrs). The obtained intervals encompass ≥95% of the values found in the control group of blood donors.

### 2.4. Poliovirus Neutralization Assays and Biosafety

PV NAb titers in serum were determined at the National Institute for Health and Welfare (THL—Helsinki, Finland). Tests were carried out according to good laboratory and microbiological practices [23] using a standardized microneutralization test for the three PV serotypes [24]. PV-susceptible HeLa cells expressing the CD155/PVR (poliovirus receptor) were cultured in 96-well microplates with 100 mL of Eagle’s essential medium supplemented with 5% FBS and antibiotic mixture. Following complement inactivation (56 °C for 30 min), serum samples were serially diluted two-fold in cell culture medium (from 1:4 to 1:8192). These dilutions were prepared in 75 µL working volumes, with 8 replicates per dilution. Then, 75 mL of medium containing 100 TCID_50_ of the appropriate PV serotype was added to each well. The virus and serum were thoroughly mixed via vortex. The microplates were incubated at 36 °C for 1 h. The content of each well (150 mL) was then transferred to a 96-well microplate containing Hela cell monolayers in 100 mL of medium. Microplates were incubated for 6 days at 36 °C in air with 5% CO_2_. The titer of PV NAb represents the highest dilution of serum capable of preventing virus-induced cytopathic effect in 4 out of 8 cell cultures tested in parallel. NAb titers were calculated using the Reed–Muench formula and are the only measurement known to correlate with protection from PV infection. Titers ≥ 1:8 are considered protective.

### 2.5. Determination of 25-Hydroxyvitamin D in Serum

Since 25(OH)D has a half-life of about 15–30 days and its levels directly reflect the body’s vitamin D stores from all sources; its levels in serum are a reliable indicator of the vitamin D status [11]. 25(OH)D was quantified by high-performance liquid chromatography–tandem mass spectrometry (HPLC-MSMS). Solvent extraction and protein precipitation of 100 µL of serum were conducted using a commercial assay from PerkinElmer (Milan, Italy), which included calibrators and quality controls. Samples (50 mL) were injected into the HPLC-MSMS system (UHPLC Agilent 1290 Infinity II, Santa Clara, CA, USA and MSMS AB-Sciex API4000, Concord, ON, Canada). Intra- and inter-assay imprecision of the assay were 4.7% and 4.3%, respectively. Sensitivity was 4 nmol/L. According to the Endocrine Society Clinical Practice Guideline on Vitamin D [25], serum 25(OH)D concentrations were interpreted as deficiency (≤50 nmol/L), insufficiency (52.5–72.5 nmol/L), and sufficiency (75–250 nmol/L).

### 2.6. Statistics

Normal distribution of values was estimated using the D’Agostino and Pearson omnibus normality test. Groups were compared using the Mann–Whitney *U* test, Fisher’s exact test, Kruskal–Wallis test, Chi-square test with Bonferroni correction (GraphPad Prism, v.10). Results are presented as median with interquartile range (IQR), mean ± SD or ±95% confidence interval (CI), geometric mean ± SD or ±95% confidence interval (CI). Two-sided *p* < 0.05 was considered statistically significant.

## 3. Results

### 3.1. Study Groups

Two hundred nine adult participants were recruited from hospitals in the Veneto and Lombardy regions, forming three distinct study groups, detailed in Table 1. Although these groups were not deliberately matched, their age, biological sex, and socioeconomic status were found to be comparable.

Among the participants, 80 polio survivors were identified as affected by PPS through comprehensive medical history review, clinical examination, and instrumental diagnostic tests. PPS cases had a median age of 56.0 years (interquartile range [IQR]: 51.8–62.0), with their diagnosis having been established at a median age of 49 years (IQR: 43.0–56.0 years). Given that the onset of Acute Paralytic Polio (APP) occurred at a median age of 2.0 years (IQR: 1.0–3.3 years), PPS typically manifested approximately 47 years after the acute disease.

The vaccination rate among PPS cases was notably low (22.1%), a finding consistent with the fact that most were born in the 1950s (before polio vaccines were widely available) and only a small proportion had been vaccinated following their acute polio episode. In contrast, vaccination rates were significantly higher in the healthy control group (100%) and among family members of PPS cases (57.7%). These divergent vaccination rates are particularly relevant for the subsequent comparison of PV antibody titers across the study groups. It should be noted that, due to limitations in sample availability, not all planned assays were performed for every study participant.

#### Immunoglobulin Levels and Deficiencies Across the Study Groups

Serum Ig levels are presented as median ± 95% confidence interval (CI). Figure 1 illustrates the results for the three major Ig classes: IgG, IgA, and IgM. Compared to healthy blood donors, IgG levels were significantly reduced by approximately 16% in both the family member group and among PPS cases. Similarly, IgA levels showed a significant reduction of about 35% in both the family member and PPS groups. In contrast, no significant differences were observed in IgM levels across any of the groups.

Further analysis of IgG subclasses, detailed in Figure 2, revealed a significant reduction in IgG1 levels among PPS cases. Family members, however, exhibited only a minor, non-significant decrease in IgG1. Relative to healthy blood donors, statistically significant reductions were observed for IgG2, IgG3, and IgG4 levels in both the family member and PPS groups.

Given that serum Ig levels were not normally distributed among subjects in the three study groups, the presence of outliers was evident. Further analysis revealed that PPS cases and their family members exhibited higher percentages of Ig levels below the lower cutoff values established from adult reference subjects (blood donors). Table 2 specifically shows that IgA, IgG, IgG1, and IgG3 exhibited the highest frequencies of concentrations below the cutoff (ranging from 10% to 34% of cases).

### 3.2. Poliovirus-Neutralizing Antibody Titers

PV NAb titers are presented as geometric mean ± 95% CI, red. Titrations were performed using eight replicates per dilution. As illustrated in Figure 3, nearly all subjects across the three study groups exhibited protective titers against all three PV types. Geometric means ranged from 1:96 to 1:384 across serotypes and groups. The maximum observed titers were 1:12,288 for both PV-1 and PV-2, and 1:3072 for PV-3.

Significant differences in PV NAb titers across groups were detected only for PV-1. Specifically, family members exhibited lower mean PV-1 titers than blood donors. Conversely, while PPS cases did not significantly differ from blood donors in PV-1 titers, they showed significantly higher titers than subjects in the family member group. For PV-2 and PV-3, NAb titers were not statistically different among groups.

The polio vaccination rate varied considerably among groups, being 100% in blood donors, 55.3% in family members, and 21.3% in PPS cases (Table 3). Consistent with these vaccination rates and other factors, non-protective PV NAb titers (<1:8) were absent in blood donors, rare among family members (2.7%), and slightly more represented among PPS cases (6.7% to 10.7%), as also shown in Table 3.

### 3.3. Serum 25-Hydroxyvitamin D Levels

Serum 25-hydroxyvitamin D [25(OH)D] concentrations are presented as median ± 95% confidence interval (CI). Measurements were performed using high-performance liquid chromatography–tandem mass spectrometry.

Figure 4 illustrates the 25(OH)D data for the three study groups. While median vitamin D levels did not significantly differ between blood donors and family members of PPS cases (85 nmol/L and 73 nmol/L, respectively), the PPS group’s median value (54 nmol/L) was significantly lower than that of both blood controls (*p* < 0.0001) and their family members (*p* < 0.0036).

Furthermore, as depicted in Figure 5, interpreting 25(OH)D concentrations according to the Endocrine Societies Guidelines [11,25] revealed that PPS cases exhibited the highest prevalence of vitamin D deficiency and insufficiency (46% and 36%, respectively). This was followed by their family members (27% deficiency, 23% insufficiency). As anticipated, healthy blood donors maintained satisfactory serum 25(OH)D levels, with only 8% presenting deficient and 10% insufficient levels.

## 4. Discussion

Our key findings indicate that a significant proportion of PPS cases exhibit reduced levels of IgG and/or IgA, supported by varying degrees of reduction in specific IgG subclasses. Thus, individuals with paralytic polio and subsequent PPS demonstrate variable types of humoral immunodeficiencies. In addition, PPS cases exhibited PV NAb titers not substantially different from those of controls but had a deficient/insufficient vitamin D status.

Notably, humoral immunodeficiencies were found to a comparable extent in their family members. This observation is particularly challenging to fully explain. While many of the family members investigated were first-degree relatives of PPS cases, approximately one-third lacked a direct genetic connection to the probands.

Evidence is being accumulating that individuals with primary immunodeficiencies are at an increased risks of vaccine-associated paralytic poliomyelitis (VAPP) and prolonged excretion of vaccine-derived PV [26]. Compared to humoral immunodeficiency patients, subjects with combined immunodeficiency show lower rates of VAPP that arises from 2 months to 4 years post-exposure to live polio vaccines [27]. The wide spectrum of Predominant Antibody Deficiencies (PADs) encompasses a group of primary disorders characterized by dysfunctional antibody production, leading to low levels of one or more Ig classes. These conditions—as reviewed by Gilchrist and Dolen [28], Griffin and Dolen [29] and others [30]—may either present with recurrent infections and reduced vaccine responses or be asymptomatic. PADs include Common Variable Immunodeficiency (CVID), which is very rare and characterized by low IgG and IgA levels; X-linked agammaglobulinemia (XLA), also very rare and marked by a near-complete absence of B cells with very low levels of all Ig classes; selective IgA deficiency, whose frequency is 1:150 to 1:2000 in Caucasians but very rare in the Japanese; IgG subclass deficiency, whose prevalence can be up to 2% of healthy individuals when considering only low levels; other less common defects. All of these lessen antiviral defenses and facilitate the establishment of chronic infections [26,31].

Overall, the observed reductions in Ig concentration, though of moderate magnitude, were statistically significant and consistently present across six different Ig classes or subclasses. This indicates a clear predisposition to serum antibody deficiencies in both the PPS group and their family members. Notably, the most pronounced deficit occurred for IgA, an antibody class involved in mucosal surface protection, but deficits were also detected for IgG, IgG1 and IgG3. Properties of IgG subclasses classes have been reviewed recently [32].

Regarding susceptibility to PVs, it would be important to assess the IgA1 and IgA2 levels in mucosal secretions of the digestive and respiratory tracts of individuals with acute paralytic poliomyelitis or PPS. This is due to dimeric IgA’s significantly higher virus-neutralizing activity compared to a corresponding IgG antibody, which confers enhanced protection [33].

IgG1 serves as the principal subclass involved in the immune response to protein antigens; therefore, its deficiency may result in inadequate antibody responses to viral agents. For instance, deficient IgG responses to the nucleocapsid and glycoprotein antigens of the Severe Fever with Thrombocytopenia Syndrome (SFTSV), a Bunyavirus, have been associated with fatal outcomes [34].

IgG3 is a uniquely effective antibody with the potential for triggering effector functions including complement activation, antibody-mediated phagocytosis, antibody-mediated cellular cytotoxicity [35]. During arboviral Chikungunya fever, the host immune response is predominantly characterized by an IgG3 response. Some patients experiencing high viremia rapidly developed robust IgG3 antibody levels. Despite enduring a more severe viremic phase, these individuals cleared the virus more rapidly, did not experience persistent arthralgia, and exhibited long-term protection [36]. Thus, IgG3 antiviral antibodies, despite their strong inflammatory potential, constitute an essential component of antiviral protection.

Therefore, given the critical importance of antibody responses against enteroviruses, PPS patients might represent individuals whose weak resistance against PVs could be complicated by the subsequent establishment of persistent infection [37].

In contrast to the observed Ig deficits in serum, PV NAb titers were substantially similar across the three study groups, an unexpected finding. When planning the study, we hypothesized that PPS cases, given their low polio vaccination prevalence, would primarily exhibit high antibody responses to a single PV serotype, i.e., the one causing the acute disease and, potentially, any subsequent persisting infection [37]. Hence, the presence of high NAb titers to heterologous PV serotypes in this group was unanticipated. Actually, fully vaccinated blood donors had the highest mean titers, followed by PPS patients and their family members. However, differences in titers were minimal and statistically significant only for PV-1. It should be noted, however, that a small proportion of PPS subjects lacked neutralizing antibodies, especially PV-2 and PV-3. These rare cases could represent individuals impacted by more severe antibody deficits.

The mean poliovirus neutralizing antibody (PV NAb) titers observed in this study are consistent with previous results obtained in adult populations from Italy [38] and Belgium [39]. This high prevalence and titer of PV antibodies (>95%) in our aged subjects can be attributed to past intensive vaccination campaigns with live virus and—in the early vaccine era—to the frequent exposure to PV carriers.

An additional consideration for interpreting this data, however, is that PPS subjects—and potentially their family members—had experienced prolonged contact with poliomyelitis patients. Specifically, following the onset of APP, in Italy severe polio cases attended specialized Rehabilitation Wards which became active from the 1950s onwards [40]. Within these environments, exposure to either wild-type or Sabin PV strains was frequent and intense.

As a final point, the prevalence of 25(OH) vitamin D deficiency was very high in PPS subjects, indicating an inadequate vitamin D status. The deficit was less pronounced in their family members and virtually absent in healthy blood donors. The results were not surprising. Individuals with PPS, though not experiencing malnutrition, are generally less exposed to open air, sunlight and physical exercise because of their clinical condition. In addition, their vitamin status indicates that they were not or inadequately supplemented with vitamin D.

Maintaining adequate vitamin D status is crucial, considering that most vitamin D is synthesized in the skin following UVB exposure, with minimal contribution from typical unfortified diets. Acknowledging the usual seasonal decline in 25(OH)D levels during winter, a target of at least 75 nmol/L is recommended for vitamin D sufficiency [11]. The inadequate vitamin D status in PPS patients is particularly concerning. Due to mobility deficits and aging, individuals with PPS frequently experience falls and often present with osteopenia/osteoporosis [41]. This combination collectively places them at a significantly increased risk for fractures. Therefore, vitamin D supplementation could confer beneficial effects [42].

Beyond its established role in bone health, vitamin D is also crucial for potentiating and regulating antimicrobial defenses [43]. This raises an intriguing, albeit difficult to assess, question: Were individuals currently diagnosed with PPS also vitamin D-deficient during infancy when APP manifested? While retrospective assessment is challenging given the currently rare incidence of new polio cases, it is known that vitamin D deficiency remains prevalent (>10%) in pediatric populations worldwide, even though most biochemically deficient children remain asymptomatic [44]. Furthermore, historical data indicate that children in countries with widespread malnutrition and compromised hygienic conditions are at a higher risk of paralytic polio [45], suggesting that complex environmental factors influence polio susceptibility.

This study presents limitations that warrant consideration when interpreting the findings. Firstly, the relatively modest sample size of the study groups may limit the statistical power and generalizability of results. Secondly, despite efforts, precise matching of cases and controls regarding age and seasonal variations was not feasible, which may have influenced immunological parameters and vitamin D levels. In fact, due to multiple factors, the age and sex of people participating in the study could not be matched as desired. A third methodological limitation was the exclusive focus on serum antibody responses. This precluded the evaluation of secretory IgA (sIgA), a key determinant of mucosal immunity and resistance against PVs.

## 5. Conclusions

Collectively, the results indicate that, despite the efficacy of current preventive vaccines, certain individuals remain particularly susceptible to the paralytogenic potential of PVs due to immunological and/or nutritional factors. This underscores the urgent need for effective PV antivirals to block possible outbreaks [46,47,48] but also—for polio survivors—the need for drugs, intravenous human immunoglobulins, and/or PV vaccines aimed at preventing or retarding the development of PPS, the latest phase of poliomyelitis [49,50].

## Figures and Tables

**Figure 1 vaccines-13-00939-f001:**
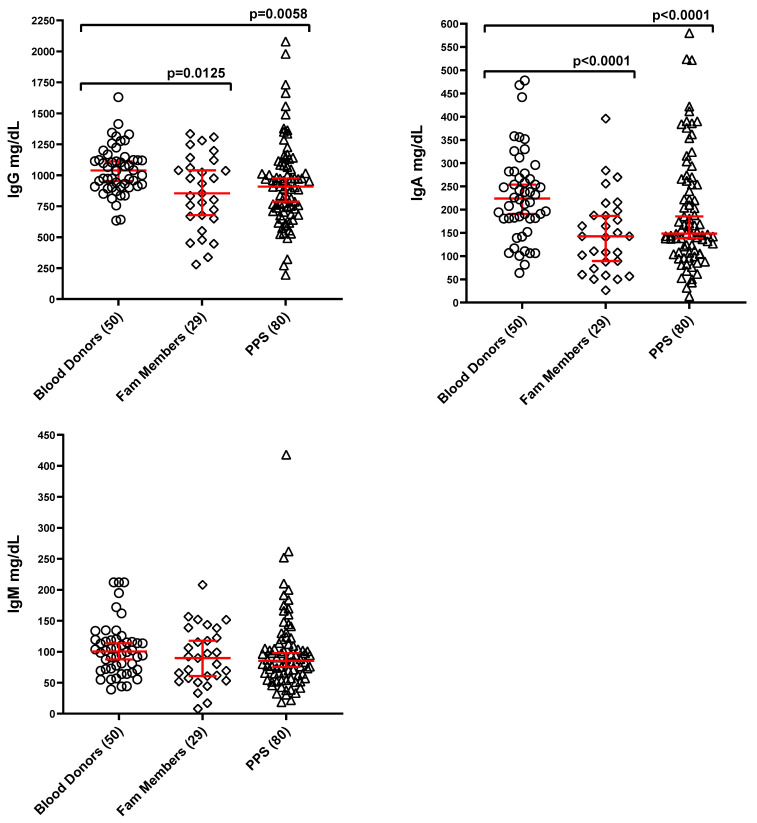
Levels of serum immunoglobulin G, A, and M in the three study groups (mg/dL; median ± 95% CI, red). Numbers of subjects in each group are reported in the X axis labeling. Because sample availability was limited, certain tests could not be conducted for all study participants. As a result, the number of subjects differs across various assays. Brackets with *p*-values indicate statistically significant differences between groups. The lack of brackets connecting two groups indicates that there are no statistically significant differences between them.

**Figure 2 vaccines-13-00939-f002:**
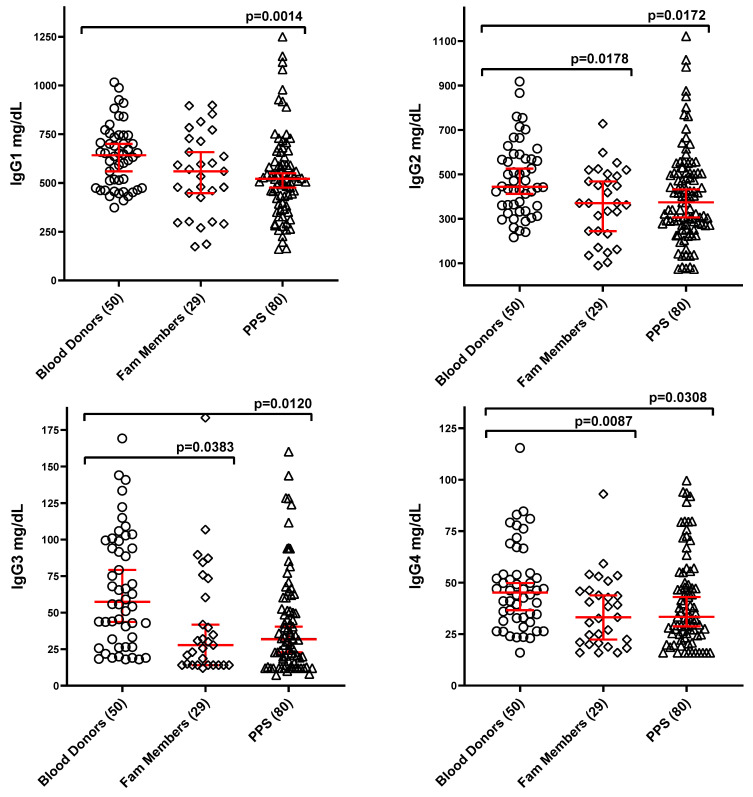
Levels of serum immunoglobulin G subclasses in the three study groups (mg/dL; median ± 95% CI, red). Numbers of subjects in each group are reported in the X axis labeling. Brackets with *p*-values indicate statistically significant differences between groups. The lack of brackets connecting two groups indicates that there are no statistically significant differences between them.

**Figure 3 vaccines-13-00939-f003:**
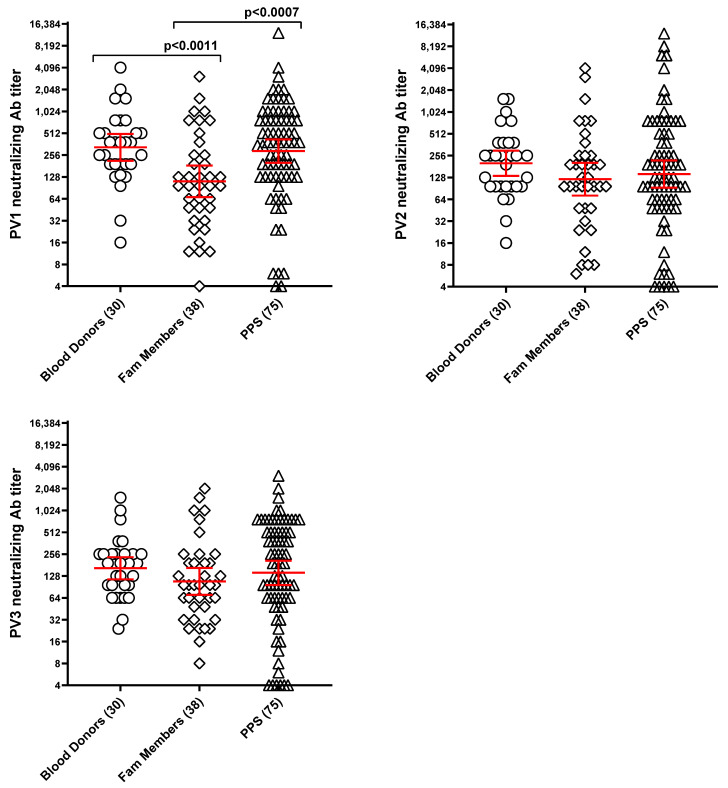
Neutralizing antibody titers to the three poliovirus serotypes in the three study groups. Titers from 1:4 to 1:16,384 are reported (geometric mean ± 95% CI, red). Numbers of subjects in each group are reported in the X axis labeling. Brackets with *p*-values indicate statistically significant differences between groups. The lack of brackets connecting two groups indicates that there are no statistically significant differences between them.

**Figure 4 vaccines-13-00939-f004:**
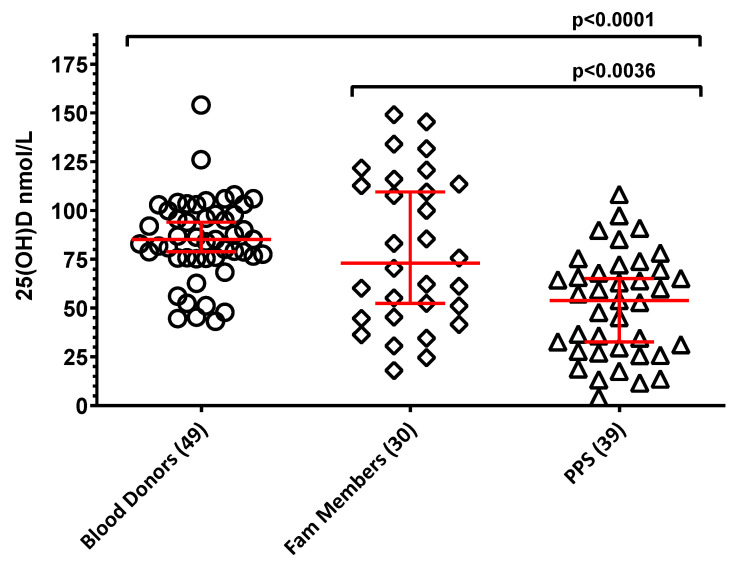
Serum 25 hydroxy vitamin D levels in the three study groups (nmol/L; median ± 95% CI, red). Numbers of subjects in each group are reported in the X axis labeling. Brackets with *p*-values indicate statistically significant differences between groups. The lack of brackets connecting two groups indicates that there are no statistically significant differences between them.

**Figure 5 vaccines-13-00939-f005:**
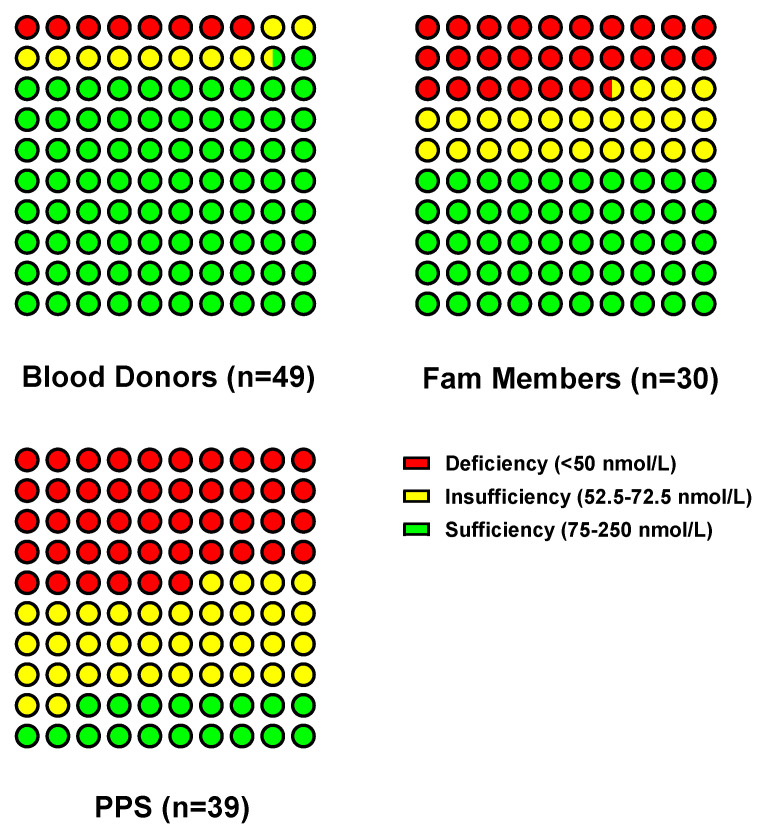
Distribution and interpretation of serum 25 hydroxy vitamin D (25(OH)D levels in healthy blood donors, family members of PPS subjects, PPS cases. Numbers of subjects in each group are reported in the X axis labeling. According to the Endocrine Society Clinical Practice Guideline on Vitamin D [25], serum 25(OH)D concentrations were interpreted as deficiency (≤50 nmol/L; red), insufficiency (52.5–72.5 nmol/L; yellow), and sufficiency (75–250 nmol/L; green).

**Table 1 vaccines-13-00939-t001:** Demographic and clinical data of study groups.

Study Groups			
Parameters	Healthy Controls (Blood Donors)	Family Members of Polio Survivors	Post-Polio Syndrome
Number of subjects	89	40	80
Percent female	46%	55.0%	57.5%
Percent polio vaccination (Salk or Sabin)	100%	57.7%	22.1%
Age (yrs) at medical visit and sample collection (median; IQR: Q1–Q3) ^1^	52.0; 42.0–58.5	49.0; 36.8–61.0	56.0; 51.8–62.0
Age (yrs) at diagnosis of APP (median; IQR: Q1–Q3) ^1^	−	−	2.0; 1.0–3.3
Age (yrs) at diagnosis of PPS (median; IQR: Q1–Q3) ^1^	−	−	49.0; 43.0–56.0

^1^ APP, Acute Paralytic Poliomyelitis; PPS, Post-Polio Syndrome; IQR, Interquartile range.

**Table 2 vaccines-13-00939-t002:** Immunoglobulin (Ig) levels lower than the cutoff values in the groups investigated.

Parameters	Healthy Controls (Blood Donors)	Family Members of Polio Survivors	Post-Polio Syndrome
Ig levels below the lower reference range in the study groups (number/total, percentage) ^1^.			
IgG < 600 mg/dL	0/50, 0%	6/29, 21%	10/80, 12%
IgA < 80 mg/dL	1/50, 2%	7/29, 24%	8/80, 10%
IgM < 40 mg/dL	1/50, 2%	3/29, 10%	7/80, 9%
IgG1 < 380 mg/dL	1/50, 0%	7/29, 24%	18/80, 22%
IgG2 < 120 mg/dL	0/50, 0%	2/29, 7%	3/80, 4%
IgG3 < 18 mg/dL	0/50, 0%	10/29, 34%	22/80, 27%
IgG4 < 18 mg/dL	1/50, 2%	3/29, 10%	10/80, 12%

^1^ Ig levels: Reference ranges for adult Caucasians have been used (Rifai 2022) [22]. Limits for significantly low Ig levels have been validated at our own laboratory (Ospedale di Circolo, Varese, Italy) using a cohort of 120 healthy adult blood donors (median age 51.2 ± 11.4 years). The obtained intervals encompass ≥95% of the values found in the reference group.

**Table 3 vaccines-13-00939-t003:** Serum titers of poliovirus-neutralizing antibodies (PV NAb) in the groups investigated.

Parameters	Healthy Controls (Blood Donors)	Family Members of Polio Survivors	Post-Polio Syndrome
Polio vaccination rate—Number/total, percentage	30/30, 100%	21/38, 55.3%	16/75, 21.3%
Non-protective serum titers (<1:8) of poliovirus-neutralizing antibodies in the study groups (number/total, percentage) ^1^:			
PV1 NAb titer < 1:8	0/30, 0%	1/38, 2.6%	5/75, 6.7%
PV2 NAb titer < 1:8	0/30, 0%	1/38, 2.6%	8/75, 10.7%
PV3 NAb titer < 1:8	0/30, 0%	0/38, 0%	7/75, 9.3%

^1^ Eight replicates per serum dilution were used for evaluating poliovirus NAb titers; titers were calculated using the Reed–Muench formula. NAb titers are the only measurement known to correlate with protection from PV infection; a titer < 1:8 is considered non-protective.

## Data Availability

Datasets are available to readers upon reasonable request to the corresponding authors.

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
