# Peer review of "Post-Polio Syndrome: Impact of Humoral Immune Deficiencies, Poliovirus Neutralizing Antibodies, Vitamin D Deficiency"

_vaccines, 2025, doi:10.3390/vaccines13090939_

Round 1

Reviewer 1 Report

Comments and Suggestions for Authors

This manuscript focuses on investigating the susceptibility factors that may lead to the development of post-polio syndrome (PPS) in elderly individuals in Italy who previously contracted polio. Although Italy has been declared polio-free due to a successful vaccination campaign, with no reported cases in recent decades, the poliovirus persists in other parts of the world. Polio remains a global health challenge, despite substantial advancements toward its eradication.

The authors’ findings reveal that, even with the availability of effective preventive vaccines, certain individuals remain highly vulnerable to the paralytic effects of poliovirus due to genetic, immunological, or nutritional factors. These results underscore the critical need for antiviral therapies to mitigate potential outbreaks, and targeted drugs and vaccines for polio survivors, aimed at preventing or delaying the onset of PPS—the final stage of polio’s long-term impact. This study offers important insights into the causes of PPS.

In general, the manuscript is well written, the information presented is well organized. I believe that this manuscript can be published in Vaccines.

Author Response

POST-POLIO SYNDROME: IMPACT OF HUMORAL IMMUNE DEFICIENCIES, POLIOVIRUS NEUTRALIZING ANTIBODIES, VITAMIN D DEFICIENCY – Toniolo A. et al.

Response to Reviewer-1

Comments and Suggestions for Authors

This manuscript focuses on investigating the susceptibility factors that may lead to the development of post-polio syndrome (PPS) in elderly individuals in Italy who previously contracted polio.

Although Italy has been declared polio-free due to a successful vaccination campaign, with no reported cases in recent decades, the poliovirus persists in other parts of the world. Polio remains a global health challenge, despite substantial advancements toward its eradication.

The authors’ findings reveal that, even with the availability of effective preventive vaccines, certain individuals remain highly vulnerable to the paralytic effects of poliovirus due to genetic, immunological, or nutritional factors. These results underscore the critical need for antiviral therapies to mitigate potential outbreaks, and targeted drugs and vaccines for polio survivors, aimed at preventing or delaying the onset of PPS—the final stage of polio’s long-term impact. This study offers important insights into the causes of PPS.

In general, the manuscript is well written, the information presented is well organized.

I believe that this manuscript can be published in Vaccines.

We are deeply grateful for your positive comments.

Reviewer 2 Report

Comments and Suggestions for Authors

Post-polio syndrome is an increasing burden among elderly survivors of polio.  While there have been past studies of antibodies in these patients, the current presentation is more comprehensive in comparing antibody levels between the patients and their family members as well as healthy controls, leading to some possible insights into an immunogenetics contribution.  Despite finding no differences in PV1 antibody titers between healthy controls and post-polio patients, the finding of significant differences in total IgGs and particularly IgA is of interest and the finding of progressively reduced vitamin D levels in family members and the patients provocative.  The IgG1 panel of Figure 2 and Figure 4 have omissions in the significance bar notations but otherwise the paper is well written and the study well executed.

Author Response

POST-POLIO SYNDROME: IMPACT OF HUMORAL IMMUNE DEFICIENCIES, POLIOVIRUS NEUTRALIZING ANTIBODIES, VITAMIN D DEFICIENCY – Toniolo A. et al.

Response to Reviewer-2

Comments and Suggestions for Authors

Post-polio syndrome is an increasing burden among elderly survivors of polio. While there have been past studies of antibodies in these patients, the current presentation is more comprehensive in comparing antibody levels between the patients and their family members as well as healthy controls, leading to some possible insights into an immunogenetics contribution.

Despite finding no differences in PV1 antibody titers between healthy controls and post-polio patients, the finding of significant differences in total IgGs and particularly IgA is of interest and the finding of progressively reduced vitamin D levels in family members and the patients provocative.

The IgG1 panel of Figure 2 and Figure 4 have omissions in the significance bar notations but otherwise the paper is well written, and the study well executed.

Thanks for your positive comments. Your question on Figures 2 and 4 is correct. Figures have been modified by removing N.S. where appropriate. For clarity, legends to Figures 1, 2, 3 and 4 have been modified as follows:

Figure 1. Levels of serum immunoglobulin G, A, and M in the three study groups (mg/dL; median ± 95% CI). Numbers of subjects in each group are reported in the X axis labeling. Because sample availability was limited, certain tests could not be conducted for all study participants. As a result, the number of subjects differs across various assays. Brackets with p-values indicate statistically significant differences between groups. The lack of brackets connecting two groups indicates that there are no statistically significant differences between them.

Figure 2. Levels of serum immunoglobulin G subclasses in the three study groups (mg/dL; median ± 95% CI). Numbers of subjects in each group are reported in the X axis labeling. Brackets with p-values indicate statistically significant differences between groups. The lack of brackets connecting two groups indicates that there are no statistically significant differences between them.

Figure 3. Neutralizing antibody titers to the three poliovirus serotypes in the three study groups. Titers from 1:4 to 1:16,384 are reported (geometric mean ± 95% CI). Numbers of subjects in each group are reported in the X axis labeling. Brackets with p-values indicate statistically significant differences between groups. The lack of brackets connecting two groups indicates that there are no statistically significant differences between them.

Figure 4. Serum 25 hydroxy vitamin D levels in the three study groups (nmol/L; median ± 95% CI). Numbers of subjects in each group are reported in the X axis labeling. Brackets with p-values indicate statistically significant differences between groups. The lack of brackets connecting two groups indicates that there are no statistically significant differences between them.

Table 3. Note 1 has been modified for clarity: 1. Eight replicates per serum dilution were used for evaluating poliovirus NAb titers; titers were calculated using the Reed-Muench formula. NAb titers are the only measurement known to correlate with protection from PV infection; a titer <1:8 is considered non-protective.

Reviewer 3 Report

Comments and Suggestions for Authors

In my opinion the provided manuscript is interesting and provides some valuable information about some factors potentially associated with Post-Polio Syndrome. I believe, the scientific soundness of the manuscript may be improved by considering the following issues:

*The inclusion criteria are not clearly described, please add

*There are no considerations about sample size. Were there any calculations made?

*Although the PPS group is described as the patients diagnosed with G14 it would be worth to know what was the clinical characteristic of that group

*The inclusion criteria for healthy controls are not known, thus the characteristic of that group is not know. Provide please who were these people, if there were patients provide their characteristics especially diagnoses. If ‘healthy’ means without any diagnosis describe, please, how these people were identified and was recruited.

*Line 200: the sentence ‘although these groups were not deliberately matched, their age, biological sex, and socioeconomic status were found to be comparable’. I do not understand how 46% of women is comparable proportion to 57.5%, or 49 yrs to 56 yrs …it is even more than 10% relative difference …

*One of the important covariate which might have an impact on a level of serum immunoglobulin is age. Consider control for age results presented on the Fig.1 and Fig.2

*Conclusions are not consistent with the aim of the study. In my opinion it should be corrected.

Author Response

POST-POLIO SYNDROME: IMPACT OF HUMORAL IMMUNE DEFICIENCIES, POLIOVIRUS NEUTRALIZING ANTIBODIES, VITAMIN D DEFICIENCY – Toniolo A. et al.

Response to Reviewer-3

Quality of English Language

( ) The English could be improved to more clearly express the research.
(x) The English is fine and does not require any improvement.

Yes

Can be improved

Must be improved

Not applicable

Does the introduction provide sufficient background and include all relevant references?

(x)

( )

( )

( )

Is the research design appropriate?

(x)

( )

( )

( )

Are the methods adequately described?

( )

(x)

( )

( )

Are the results clearly presented?

( )

(x)

( )

( )

Are the conclusions supported by the results?

( )

( )

(x)

( )

Are all figures and tables clear and well-presented?

(x)

( )

( )

( )

Comments and Suggestions for Authors

In my opinion the provided manuscript is interesting and provides some valuable information about some factors potentially associated with Post-Polio Syndrome.

I believe, the scientific soundness of the manuscript may be improved by considering the following issues:

  1. *The inclusion criteria are not clearly described, please add.

Thanks for the request. Inclusion criteria have been included as follows:

Lines 111-120: Polio survivors were admitted to participating Hospitals because of progressive worsening of weakness and fatigue during daily and/or working activities. According to principles of the European Federation of Neurological Societies [20], polio survivors were classified as having PPS if they met the following criteria: a) confirmed history of polio; b) a period of partial or fairly complete functional recovery after the acute episode; c) a period of at least 15 years of stable neurological function; d) two or more of the following new symptoms after the stable period: new muscle weakness, muscle fatigue, new muscle atrophy, muscle and joint pain, breathing or swallowing problems, sleep disorders, cold intolerance; e) exclusion of other neurological, medical and orthopedic problems. PPS is characterized by gradual progression over time [20,21]. Clinical features include new muscular weaknesses persisting for at least one year, fasciculations, atrophy of previously unaffected limbs, functional deficit of bulbar-innervated muscles, muscles of respiration, chronic fatigue, joint and muscle pain, cold intolerance, swallowing problems, sleep disorders, a reactive depressive syndrome.

Lines 102-110: Control groups comprised: a) healthy family members of PPS subjects who had lived with them for at least five years (64% were siblings and offspring, 36% were partners or kindred). The clinical conditions of family members were evaluated by the same medical staff who authored this manuscript; and b) healthy controls (blood donors of 35 to 60 years of age). With some exceptions, in Italy people can donate blood from age 18 to 60. Blood samples from each PPS subject, his/her family member(s) and control blood donors were obtained on the same day. The number of subjects in each of the three groups was determined by the number of serum samples that were available for each assay. At the time the study was designed, it was therefore impossible to predict precisely the number of cases and controls. Nonetheless, the study's initial requirement of at least 28 participants per group was met for all assays.

  1. *There are no considerations about sample size. Were there any calculations made?

Lines 107-110: The number of subjects in each of the three groups was determined by the number of serum samples available for each assay. At the time the study was designed, it was impossible to predict the number of recruitable cases and controls. Nonetheless, the study's initial requirement of at least 28 participants per group was met for all assays.

3-           Although the PPS group is defined as patients diagnosed with G14, it would be valuable to provide information regarding the clinical characteristics of this group.

The question is addressed in point-1, Lines 111-120.

  • *The inclusion criteria for healthy controls are not known, thus the characteristic of that group is not known. Provide please who these people were, if there were patients provide their characteristics especially diagnoses. If “healthy” means without any diagnosis describe, please, how these people were identified and were recruited.

The question is addressed in point-1, lines 102-107: Control groups comprised: a) healthy family members of PPS subjects who had lived with them for at least five years (64% were siblings and offspring, 36% were partners or kindred). The clinical conditions of family members were evaluated by the same medical staff who authored this manuscript; and b) healthy controls (blood donors of 35 to 60 years of age – with some exceptions, in Italy people can donate blood from age 18 to 60). Blood samples from each PPS subject, his/her family member(s) and control blood donors were obtained on the same day.

  • Line 200: the sentence ‘although these groups were not deliberately matched, their age, biological sex, and socioeconomic status were found to be comparable’. I do not understand how 46% of women is comparable proportion to 57.5%, or 49 yrs to 56 yrs …it is even more than 10% relative difference…

You are right, demographic differences among groups are in the range 10-15% which may not be adequate for a classic case-control study.

In the case of PPS patients (the main group to be investigated) the number of subjects to recruit was difficult to predict, since – when the study was designed – the percentage of Italian polio survivors who were to develop PPS was unknown (estimates varied form 10 to 80%). Similarly, the number of heathy family members living with PPS subjects for at least five years and consenting to participate in the study could not be predicted. Once more, the number of healthy blood donors in the age range 35 to 60 years was difficult to predict, due to the fact that blood samples had to be taken on the same day when PPS cases presented themselves at participating Hospitals. The Transfusion Center at the Hospital was processing about 100 donations per day, but the most frequent age of donors was 18 to 40 years. People over 40 years were rare and a minority of them consented to participate in the study. Due to the multiple factors above, the age and sex of people participating in the study could not be matched as desired (Lines 329-330). This further limitation of our study is noted in the Discussion section. However, comparing the available case and control groups yielded biologically meaningful results. We are confident these findings may help inform the design of future studies on immunodeficiencies and nutritional deficits in people with PPS.

  1. One of the important covariates which might have an impact on a level of serum immunoglobulin is age. Consider control for age results presented in Fig.1 and Fig.2.

This is also a correct notation on age and variation of Ig levels in serum. However, the data reported here is not intended as a reference for immunoglobulin levels in PPS patients in general. The data indicate, however, that significant differences have been noted in Italy between PPS cases and healthy controls in populations of “similar” age and sex. This point will help inform the design of possible immunodeficiencies in people with paralytic poliomyelitis and/or PPS.

  1. Conclusions are not consistent with the aim of the study. In my opinion it should be corrected.

We appreciate the prudent notation of the Reviewer. However, it is recognized that only a small minority of subjects infected by polioviruses proceeds to manifest a paralytic syndrome. Even within the same family there are notable differences in susceptibility to PVs. The need for antiviral drugs is supported by WHO (Badizadegan K, Kalkowska DA, Thompson KM. Health Economic Analysis of Antiviral Drugs in the Global Polio Eradication Endgame. Med Decis Making. 2023 Oct-Nov;43(7-8):850-862) and the search for drugs (or immunological interventions) is recommended by physical therapists and neurologists. An article by our group on the etiology/pathogenesis of PPS that justifies the need for preventive treatments has been just accepted by J Neurology. The paper is currently in Press. With the permission of the Editor, the latter reference may be inserted in the present article during the publication process.

Lines 335-339: text has been modified as follows (with the addition of one reference): Collectively, the results indicate that, despite the efficacy of current preventive vaccines, certain individuals remain particularly susceptible to the paralytogenic potential of PVs due to immunological and/or nutritional factors. This underscores the urgent need for effective PV antivirals to block possible outbreaks [46–48] but also - for polio survivors – the need for drugs and PV vaccines aimed at preventing or retarding the development of PPS, the latest phase of poliomyelitis [49,50].

Submission date 21 July 2025

Date of this review 15 Aug 2025

Date of Response to Reviewers 27 Aug 2025
